# Factors associated with interstitial lung disease in patients with rheumatoid arthritis: A systematic review and meta-analysis

**Minjie Zhang[1], Jianwei Yin[2]\*, Xiaoyan Zhang[3]**

**1** Department of Medical Laboratory, Xian Yang Central Hospital, Xianyang, China, **2** Department of Respiratory and Critical Care Medicine, Yulin No. 2 Hospital, Yulin, China, **3** Department of Medical Laboratory, Yan'an People's Hospital, Yan'an, China

\* maljyanan@163.com

## Abstract

### Objectives

Interstitial lung disease (ILD) is frequent in patients with rheumatoid arthritis (RA) and is a potentially life-threatening complication with significant morbidity and mortality. This meta-analysis aims to systematically determine the factors associated with the development of rheumatoid arthritis–related interstitial lung disease (RA-ILD).

### Materials and methods

All primary studies which reported the factors associated with of RA-ILD were eligible for the review except case reports. The Cochrane Library, PubMed, Embase, Web of Science, Chinese Biological Medicine Database (CBM), China National Knowledge Infrastructure (CNKI), and WANFANG electronic databases were searched through to December 30, 2022, for studies investigating the factors associated with RA-ILD. The methodological quality assessment of the eligible studies was performed using the Newcastle-Ottawa Scale (NOS). 2 reviewers extracted relevant data independently. Then, weighed mean differences (WMDs) or pooled odds ratios (ORs) and corresponding 95% confidence intervals (CIs) were obtained for the relationships between the factors and RA-ILD. The statistical meta-analysis, subgroup and sensitivity analyses were performed using the Review Manager 5.3, and publication bias with Egger's test were performed using the Stata12.0 software.

### Results

A total of 22 articles were screened for a meta-analysis which involved 1887 RA-ILD patients and 8066 RA without ILD patients. Some identified factors that were associated with an increased risk of RA-ILD included male sex (OR = 1.92, 95% CI: 1.54–2.39; $P <$ 0.00001), older age (WMD = 5.77 years, 95% CI: 3.50–8.04; $P <$ 0.00001), longer duration of RA (WMD = 0.80 years, 95% CI 0.12–1.47; $P =$ 0.02), older age at onset of RA (WMD = 6.41 years, 95% CI: 3.17–9.64; $P =$ 0.0001), smoking (OR = 1.69, 95% CI: 1.30–2.18; $P <$ 0.0001). Five factors of laboratory items associated with the development of RA-ILD were

**Data Availability Statement:** All relevant data are within the paper and its Supporting Information files.

**Funding:** The author(s) received no specific funding for this work.

**Competing interests:** The authors have declared that no competing interests exist.

**Abbreviations:** ILD, Interstitial lung disease; RA, rheumatoid arthritis; RA-ILD, rheumatoid arthritis–related interstitial lung disease; CBM, Chinese Biological Medicine Database; CNKI, China National Knowledge Infrastructure; NOS, Newcastle-Ottawa Scale; WMD, weighed mean difference; OR, odds ratio; CI, confidence intervals; RF, rheumatoid factor; ACPA, anti-citrullinated protein antibodies; ESR, erythrocyte sedimentation rate; CRP, C-reactive protein; ANA, antinuclear antibody; HRCT, high-resolution computed tomography.

evaluated in the meta-analysis. Compared with RA without ILD patients, positive rheumatoid factor (RF) (OR = 1.72, 95% CI: 1.47–2.01; $P < 0.00001$) and positive anti-citrullinated protein antibodies (ACPA) (OR = 1.58, 95% CI: 1.31–1.90; $P < 0.00001$) increased the risk of RA-ILD. Meanwhile, RF titer (WMD = 183.62 (IU/mL), 95% CI: 66.94–300.30; $P = 0.002$) and ACPA titer (WMD = 194.18 (IU/mL), 95% CI: 115.89–272.47; $P < 0.00001$) were significantly associated with increased risk of RA-ILD. Elevated erythrocyte sedimentation rate (ESR) (WMD = 7.41 (mm/h), 95% CI: 2.21–12.61; $P = 0.005$) and C-reactive protein (CRP) (WMD = 4.98 (mg/L), 95% CI: 0.76–9.20; $P = 0.02$) were also significantly associated with the development of the RA-ILD, whereas antinuclear antibody (ANA) positive status was not significantly associated with increased risk of RA-ILD (OR = 1.27, 95% CI: 1.00–1.60; $P = 0.05$).

## Conclusions

This meta-analysis showed that male gender, older age, longer duration of RA, older age at onset of RA, smoking, positive RF, positive ACPA, elevated RF titer, elevated ACPA titer, higher ESR and higher CRP were associated with RA-ILD.

## Introduction

Rheumatoid arthritis (RA) is a systemic autoimmune disease characterized by chronic inflammation of the joints that eventually lead to symmetric arthritis and bony destruction [1]. The lungs are a normally affected extraarticular location, and it can involve the pleura, airways, parenchyma, and vasculature [2]. Parenchymal disease in the form of interstitial lung disease (ILD) is a known complication with a potentially risk of devastating manifestation [3], however, reported prevalence and incidence of rheumatoid arthritis-related interstitial lung disease (RA-ILD) is variable, depending on the heterogeneous clinical presentation, disease course, the eligibility diagnostic criteria and the definition of disease types. Although the definition of ILD is based on histopathological criteria, currently, the diagnosis is established by high-resolution computed tomography (HRCT) and can often be established based on pulmonary function and clinical symptoms [4]. There are no formal guidelines for clinical screening of ILD in RA patients and specific strategic therapy has not been established yet. To identify factors/ predictors associated with ILD in people with RA is crucial to better understand the clinical characteristics and allow for earlier diagnosis and better treatment with the goal of preventing irreversible lung damage [5]. To date, there have not been any guidelines or RCTs for the treatment of RA-ILD. Although a great deal of research has found that the treatment of RA played beneficial roles in the prevention of RA-ILD. There are still many questions about RA-ILD. Some studies have tried to identify risk factors for RA-ILD, including demographic factors, such as male sex, age, duration of RA, age at onset of RA and smoking status. However, there is a dearth of specific predictive biomarkers to predict the development of RA-ILD at present. Few studies have investigated the association between serological factors on RA-ILD risk. It has been reported that an increased rheumatoid factor (RF), and anti-citrullinated protein antibodies (ACPA) antibody titer are predictors for the development of RA-ILD [6,7]. In addition, several other serological factors associated with RA-ILD have also been described, including antinuclear antibody (ANA), elevated erythrocyte sedimentation rate (ESR), and C-reactive protein (CRP). Awareness and knowledge of factors associated with RA-ILD are

crucial to the prevention of irreversible lung injury by early detection. Recently, while many studies have been published to assess the demographic and serological features of RA-ILD patients in comparison to RA patients without ILD and to identify the factors associated with ILD, the results are discordant, a few have not found an association. This systematic review and meta-analysis were performed to identify factors associated with RA-ILD.

## Materials and methods

### Search strategy

We searched the Chinese Biological Medicine Database (CBM), China National Knowledge Infrastructure (CNKI), Cochrane Library, Medline via PubMed, Embase, Web of Science, and WANFANG electronic databases. Searches were restricted to articles written in English and Chinese. We searched all relevant studies on RA-ILD published before December 30, 2022 and two authors extracted data from articles independently. The following keywords, subject headings and text words were used: "rheumatoid arthritis" OR "arthritis" OR "rheumatoid" OR "RA" combined with "interstitial lung disease" OR "lung diseases" OR "ILD". In addition, relevant references of the articles identified through the search process were screened to identify potential primary articles.

### Inclusion and exclusion criteria

All studies had to meet the following eligibility criteria: (1) studies reporting or providing data investigating risk factors for RA-ILD; (2) studies with detailed diagnostic criteria information about the ILD (including findings on HRCT, CT, clinical symptoms, chest x-ray, lung biopsy, pulmonary function tests, dyspnea scale, radiographic evidence) in RA-ILD patients; (3) included cases in accordance with definitive diagnosis criteria of RA; (4) included at least one potential risk factor. (5) included sufficient information to calculate weighed mean differences (WMDs) with 95% confidence intervals (CIs) and pooled odd ratios (ORs) with 95% CIs; (6) Only articles published in English and Chinese were considered.

The exclusion criteria of this study were set as follows: (1) lacked a control group (lacking RA without ILD patients) or provided data by comparing the difference in ILD between RA-ILD; (2) studies reporting only the prevalence, survival and characteristics of RA-ILD; (3) studies reporting the risk factors for progression and prognosis of RA-ILD; (4) the studies whose results were not presented as mean ± standard deviation or number (%); (5) RA patients in control group were complicated with other autoimmune diseases; (6) data could not be extracted.

### Data extraction and quality assessment

Data from included studies were extracted by 2 researchers independently using the standard data collection forms. The following data was extracted from each eligible study: name of the first author, year of publication, study design, RA classification criteria, ILD diagnosis methods, the number of participants and their demographic features such as the gender, average age, duration of RA, age at onset of RA. In addition, the methodological quality of cross-sectional studies was evaluated using the assessment involving 11 items recommended by the Agency for Healthcare Research and Quality (AHRQ). The total score ranged from 0 to 11, with a score of 8 or higher considered high quality. The methodological quality of cohort and case-control studies were assessed using the Newcastle-Ottawa scale (NOS) with a total score of 9 points. A total score of 5 or less was considered low, 6 or 7was considered moderate, and 8 or 9 was deemed high quality. four of the studies were considered low quality, sixteen were

considered moderate quality, and two were determined to be high quality based on their total scores. Evaluation of the quality of included studies was performed by 2 authors independently based on the study design. Two independent authors were blinded to the authors names, titles and years of publication of the included studies. Discrepancies in scores were resolved by consensus with a third author. The methodological quality assessment results were shown in S2 Table.

## Statistical analysis

The Review Manager 5.3 software was used to calculate the summary values for estimating factors associated with RA-ILD. The results were presented as pooled ORs with 95% CIs for categorical variables and WMDs with 95% CIs for continuous variables. Heterogeneity test between studies was assessed using inconsistency index ($^2$) statistic and Cochran-Q statistic. A *P* value of <0.05 for the Cochrane Q test was considered to indicate significant heterogeneity, then the random effects model was used for the analysis among the included studies. The results from the fixed-effect model were presented only when there was mild and statistically non-significant heterogeneity. Subgroup analysis was performed to investigate the impact of different factors on the heterogeneity between the included studies and sensitivity analysis was conducted to determine the impact of each individual studies on the heterogeneity by sequential omission of individual studies. Forest plots were used to display the results from the individual studies and the pooled estimates. The potential for publication bias was evaluated by Egger test using STATA software (version 12). A *P* value of <0.05 was considered statistically significant.

## Results

### Study characteristics

The search of the CBM, CNKI, Cochrane Library, Medline via PubMed, Embase, Web of Science, and WANFANG electronic databases up to December 30, 2022. A total of 3942 literatures were obtained through preliminary screening, of which 1861 remained after duplicates were removed. 1814 articles were excluded after title screening and evaluating abstracts. The full text of the remaining 47 articles were reviewed for eligibility. Of these, 25 articles which were not related to the risk of developing RA-ILD or did not meet the eligibility criteria were discarded. Finally, 22 articles were included [8–29]. A flowchart of article screening was showed in Fig 1.

### Data synthesis and meta-analysis

S1 Table list the characteristics of the included 22 studies with a total of 1887 RA-ILD patients and 8066 RA without ILD patients, including 15 retrospective cohort studies, and 7 observational studies (4 case-control studies and 3 cross-sectional studies). The pooled analysis results of the potential factors were as follows.

Demographic characteristics: analysis of 1858 RA-ILD patients and 7944 RA without ILD patients showed that male was associated with increased risk of RA-ILD (OR = 1.92, 95% CI: 1.54–2.39; *P* < 0.00001) in 21 studies (Fig 2). Number of males was not available for RA-ILD patients and RA without ILD patients in "Salaffi 2019" study. The identified factors associated with RA-ILD include average age, duration of RA, age at onset of RA and smoking. The meta-analysis results of the average age within the 16 studies reporting WMD age (WMD = 5.77 years, 95% CI: 3.50–8.04; *P* < 0.00001) (Fig 3) revealed that older age was associated with RA-ILD. Furthermore, RA-ILD patients had a longer RA duration than RA without ILD

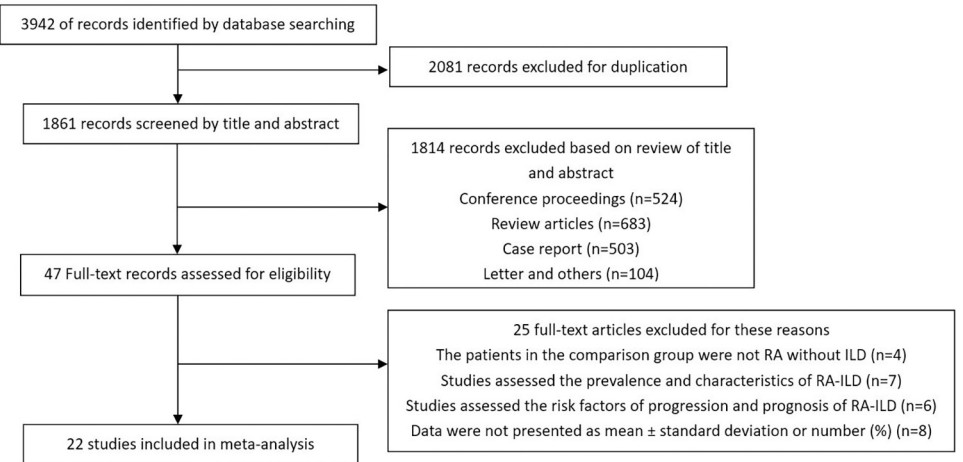

**Fig 1. Flow diagram of selecting the literature and screening process.**

patients (WMD = 0.80 years, 95% CI 0.12–1.47; *P* = 0.02) (Fig 4). In addition, the risk of RA-ILD was increased in patients who were older at onset of RA (WMD = 6.41 years, 95% CI: 3.17–9.64; *P* = 0.0001) (Fig 5). A subgroup analysis by smoking status (divided into "ever-smoking group" and "current smoker group") was performed to determine the association of smoking with RA-ILD, the pooled analysis results suggested that either smoking history (OR = 1.51, 95% CI: 1.08–2.13; *P* = 0.02) or current smokers (OR = 2.03, 95% CI: 1.37–3.01; *P* = 0.0004) had increased risk for RA-ILD (Fig 6).

## There were ever and current smoking datas included in kodrui 2010 study and Yang 2019 study

Laboratory data: a total of 5 laboratory items were identified as factors associated with the RA-ILD. These were RF, ACPA, ANA, ESR, and CRP, we evaluated all of these factors for RA-ILD. The results of analysis showed positive RF (OR = 1.72, 95% CI: 1.47–2.01;

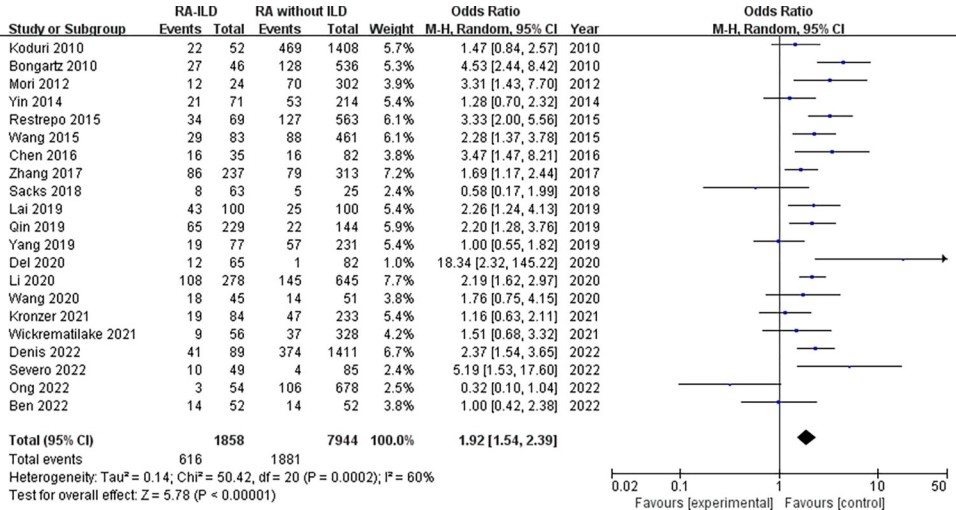

**Fig 2. Forest plot of the pooled ORs for correlation of male sex with RA-ILD.**

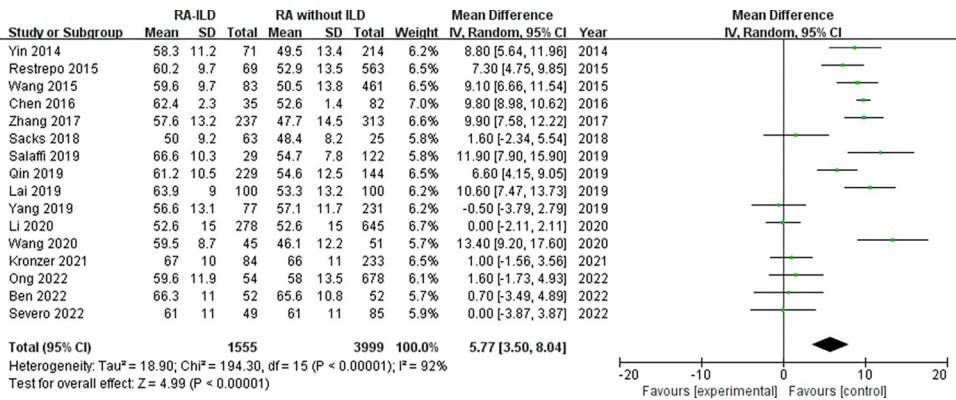

**Fig 3. Forest plot of the pooled WMDs about average age.**

$P < 0.00001$) (Fig 7), positive ACPA (OR = 1.58, 95% CI: 1.31–1.90; $P < 0.00001$) (Fig 8), RF titer (WMD = 183.62 (IU/mL), 95% CI: 66.94–300.30; $P = 0.002$) (Fig 9), ACPA titer (WMD = 194.18 (IU/mL), 95% CI: 115.89–272.47; $P < 0.00001$) (Fig 10), ESR level (WMD = 7.41 (mm/h), 95% CI: 2.21–12.61; $P = 0.005$) (Fig 11), and CRP level (WMD = 4.98 (mg/L), 95% CI: 0.76–9.20; $P = 0.02$) (Fig 12) were associated with increased risk of RA-ILD. However, positive ANA (OR = 1.27, 95% CI: 1.00–1.60; $P = 0.05$) (Fig 13) was not significantly associated with increased risk of RA-ILD.

## Heterogeneity test and sensitivity analysis

Significant heterogeneity was observed for smoking, RF titer, ACPA titer, and ESR level (Table 1). In order to identify possible sources of heterogeneity, sensitivity analysis was conducted. The analyses were repeated by removing one study per iteration by using Review Manager. No single study significantly affected the pooled effects. Nevertheless, no significant heterogeneity was observed for RF titer (chi$^2$ = 2.01, $P = 0.57$, I$^2$ = 0%; fixed effects model) and ESR level (chi$^2$ = 6.87, $P = 0.23$, I$^2$ = 27%; fixed effects model) when study "Salaffi 2019" was removed as well as for ACPA titer (chi$^2$ = 8.40, $P = 0.08$, I$^2$ = 52%; fixed effects model) when study "Wang 2020" was removed.

## Subgroup analysis

Due to significant heterogeneity for the analysis of smoking, ACPA titer and RA-ILD risk, subgroup analysis was conducted to evaluate the source of heterogeneity between the studies. Subgroup analysis by smoking status (divided into "ever-smoking group" and "current smoker

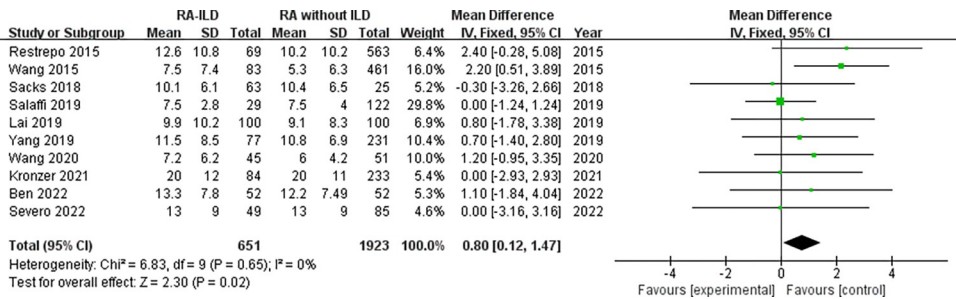

**Fig 4. Forest plot of the pooled WMDs about duration of RA.**

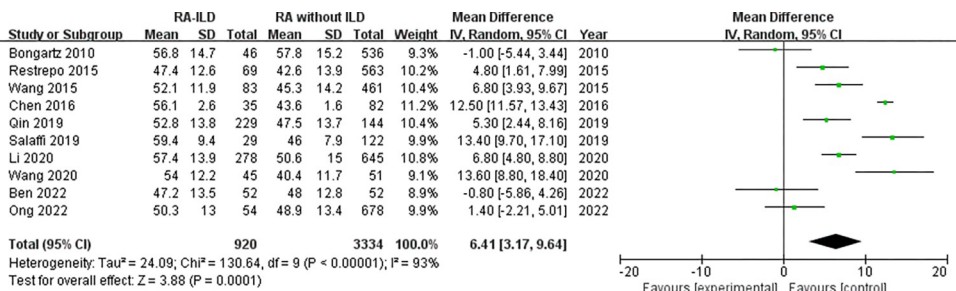

**Fig 5. Forest plot of the pooled WMDs about age at onset of RA.**

group") revealed that ever-smoking group (chi$^2$ = 29.70, $P$ = 0.001, I$^2$ = 66%; random effects model) had a higher heterogeneity compared to current smoker group (chi$^2$ = 9.85, $P$ = 0.13, I$^2$ = 39%; random effects model). There was no significant heterogeneity in the subgroup of current smoker (Fig 6). Based on the study area, subgroup analysis for the ACPA titer by region (divided into "foreign group" and "China group") revealed that ACPA titer both in foreign and China populations was significantly related to the risk of RA-ILD, while there was no heterogeneity in the subgroup of foreign group (chi$^2$ = 1.20, $P$ = 0.55, I$^2$ = 0%; random effects model). In addition, the ACPA titer of RA-ILD in China group (WMD = 380.30 (IU/mL), 95% CI: 137.31–623.29; $P$ = 0.002) was higher than that of foreign group (WMD = 136.99 (IU/mL), 95% CI: 101.64–172.34; $P$ < 0.00001) (Fig 14).

## Risk of bias assessment

Publication bias of the included studies was examined. No significant publication bias was found by using Egger's tests except for the analysis of age at onset of RA ($t$ = -3.10, $P$ = 0.015),

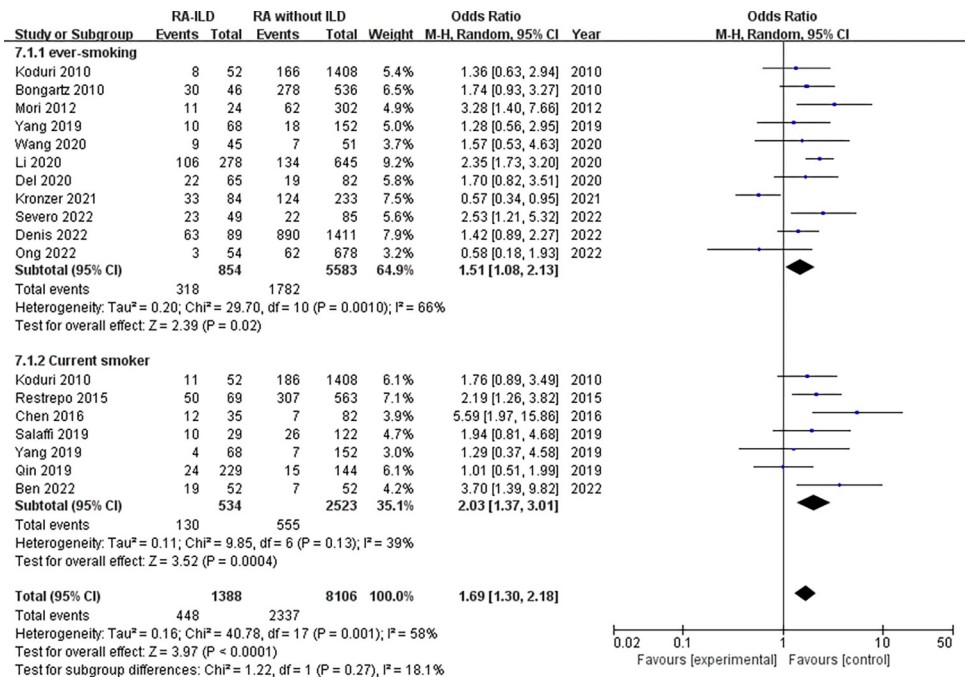

**Fig 6. Forest plot of the pooled ORs for correlation of smoking with RA-ILD.**

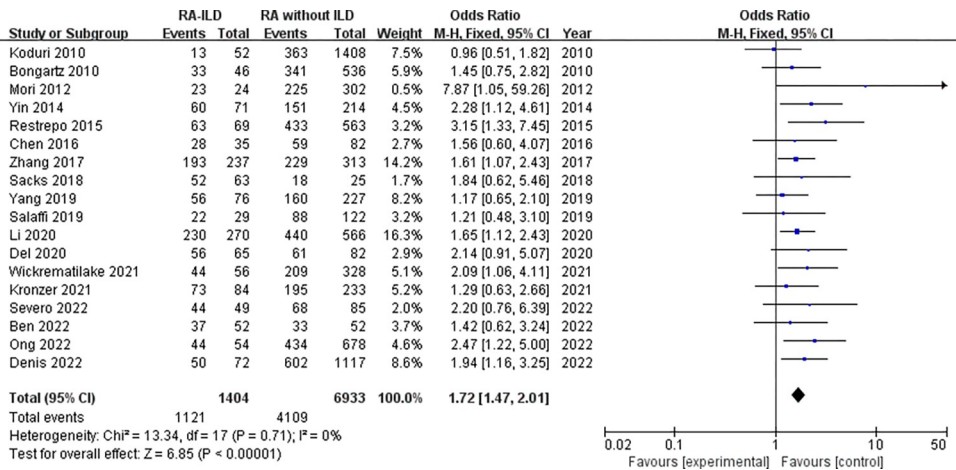

**Fig 7. Forest plot of the pooled ORs for correlation of positive RF with RA-ILD.** Data about positive RF was available for 72 RA-ILD patients and 1117 RA without ILD patients in Denis 2022 study.

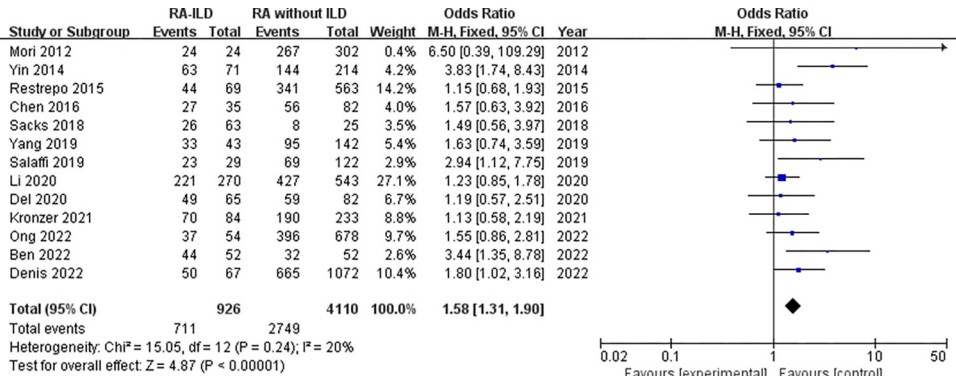

**Fig 8. Forest plot of the pooled ORs for correlation of positive ACPA with RA-ILD.** Data about positive ACPA was available for 67 RA-ILD patients and 1072 RA without ILD patients in Denis 2022 study.

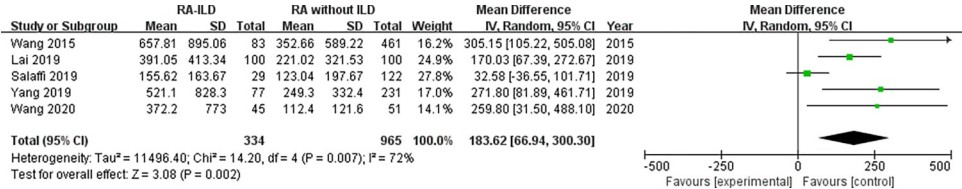

**Fig 9.** Forest plot of the pooled WMDs for correlation of RF titer with RA-ILD.

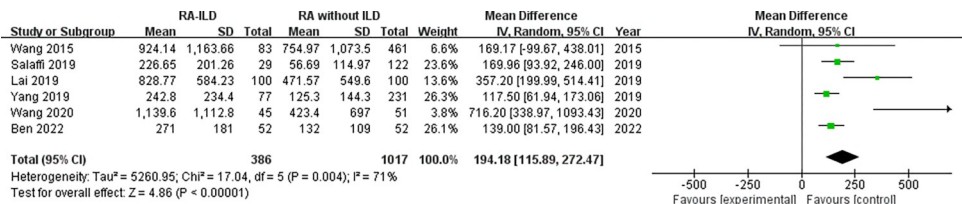

**Fig 10. Forest plot of the pooled WMDs for correlation of ACPA titer with RA-ILD.**

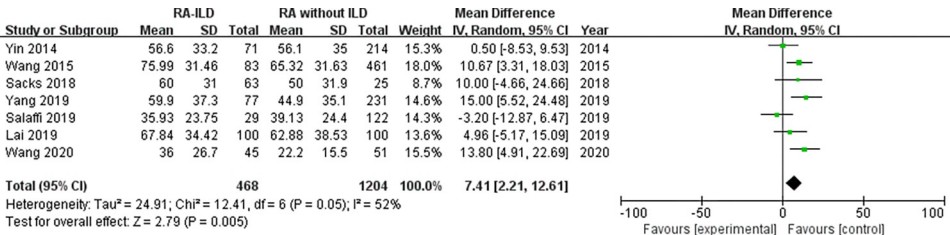

**Fig 11. Forest plot of the pooled WMDs for correlation of ESR level with RA-ILD.**

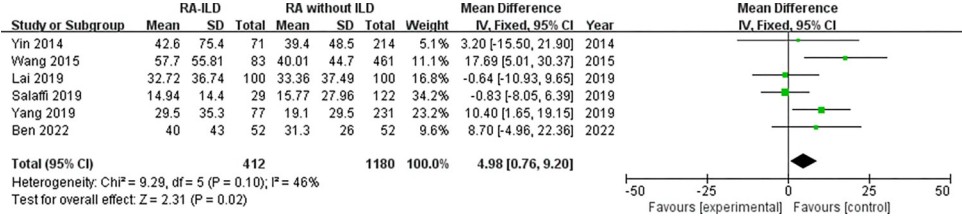

**Fig 12. Forest plot of the pooled WMDs for correlation of CRP level with RA-ILD.**

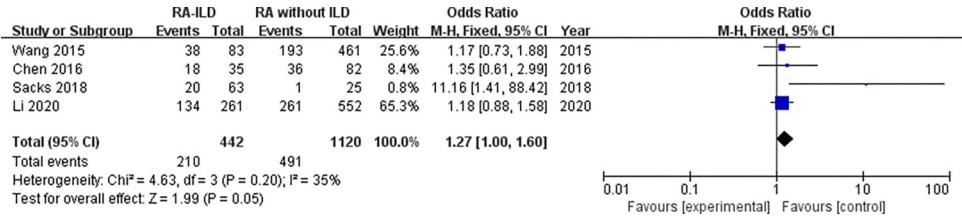

**Fig 13. Forest plot of the pooled ORs for correlation of positive ANA with RA-ILD.**

positive ACPA ($t = 2.61$, $P = 0.024$), RF titer ($t = 4.27$, $P = 0.024$) and ACPA titer ($t = 2.89$, $P = 0.044$) (Table 1).

## Discussion

In this systematic review and meta-analysis, we aimed to identify the factors associated with the development of RA-ILD. The results demonstrated that RA-ILD patients were older than the RA without ILD patients. Older patients might have a longer disease duration, and we found that longer duration of RA was associated with RA-ILD. Meanwhile, we found that ILD is more common in patients who were older at onset of RA. Thus, older age might be a useful predictor for predicting increased risk of developing RA-ILD and it is essential to screen ILD in older RA patients. Male sex was also associated with RA-ILD even though RA is more common among females [30]. Furthermore, smoking was associated with a substantially increased risk of RA-ILD. The mechanism for the association of smoking with RA-ILD remains unclear. However, it is well known that cigarette smoke components could trigger an immune reaction, and the production of serum autoantibodies against multiple citrullinated proteins in the lung may result in inflammation and epithelial cell injury which finally leads to ILD [31]. Given the heterogeneity in identifying smoking, subgroup analysis was conducted. Both current smoker and ever-smoking were statistically associated with an increased risk of RA-ILD, there was also no a significantly heterogeneity between current smoker and ever-smoking ($chi^2 = 1.22$,

**Table 1. Factors associated with RA-ILD.**

| Factors | Studies (n) | Pooled effect | | Heterogeneity | | Egger's test | |
|---|---|---|---|---|---|---|---|
| | | OR/WMD [95%CI] | P | I$^2$ (%) | P | t | P |
| Demographic | | | | | | | |
| Male | 21 | 1.92 (1.54–2.39) | < 0.01 | 60 | < 0.01 | -0.34 | 0.739 |
| Average age | 16 | 5.77 (3.50–8.04) | < 0.01 | 92 | < 0.01 | -2.09 | 0.056 |
| Duration of RA | 10 | 0.80 (0.12–1.47) | 0.02 | 0 | 0.65 | 0.30 | 0.772 |
| Age at onset of RA | 10 | 6.41 (3.17–9.64) | < 0.01 | 93 | < 0.01 | -3.10 | 0.015[a] |
| Smoking | 16 | 1.69 (1.30–2.18) | < 0.01 | 58 | < 0.01 | -0.16 | 0.876 |
| Ever-smoking (Subgroup) | 11 | 1.51 (1.08–2.13) | 0.02 | 66 | < 0.01 | -0.77 | 0.461 |
| Current smokers (Subgroup) | 7 | 2.03 (1.37–3.01) | < 0.01 | 39 | 0.13 | 0.73 | 0.498 |
| Laboratory data | | | | | | | |
| Positive RF | 18 | 1.72 (1.47–2.01) | < 0.01 | 0 | 0.71 | 1.41 | 0.177 |
| Positive ACPA | 13 | 1.58 (1.31–1.90) | < 0.01 | 20 | 0.24 | 2.61 | 0.024[a] |
| RF titer | 5 | 183.62 (66.94–300.30) | < 0.01 | 72 | < 0.01 | 4.27 | 0.024[a] |
| RF titer (omitting Salaffi 2019) | 4 | 217.52 (140.11–294.94) | < 0.01 | 0 | 0.57 | 3.80 | 0.063 |
| ACPA titer | 6 | 194.18 (115.89–272.47) | < 0.01 | 71 | < 0.01 | 2.89 | 0.044[a] |
| ACPA titer (omitting Wang 2020) | 5 | 147.94 (113.73–182.15) | < 0.01 | 52 | 0.08 | 1.62 | 0.204 |
| ACPA titer foreign group (Subgroup) | 3 | 136.99 (101.64–172.34) | < 0.01 | 0 | 0.55 | 2.36 | 0.255 |
| ACPA titer China group (Subgroup) | 3 | 380.30 (137.31–623.29) | < 0.01 | 63 | 0.07 | 0.41 | 0.751 |
| ESR level | 7 | 7.41 (2.21–12.61) | < 0.01 | 52 | 0.05 | -0.29 | 0.781 |
| ESR Level (omitting Salaffi 2019) | 6 | 9.28 (5.47–13.09) | < 0.01 | 27 | 0.23 | -0.16 | 0.878 |
| CRP level | 6 | 4.98 (0.76–9.20) | 0.02 | 46 | 0.10 | 0.89 | 0.424 |
| Positive ANA | 4 | 1.27 (1.00–1.60) | 0.05 | 35 | 0.20 | 2.40 | 0.138 |

a: P < 0.05, significant publication bias was observed.

P = 0.27, I$^2$ = 18.1%). Additionally, heterogeneity in the subgroup of current smoker was low. Together, these results demonstrated the importance of smoking cessation to prevent RA-ILD, especially in patients who were a current smoker.

We found a considerable positive RF and ACPA as well as a higher titer of RF and ACPA in RA-ILD patients. Both of these antibodies were significantly related to the risk of RA-ILD at the high titer, which were customarily done in clinical laboratories highlighting the potential importance of these antibodies in the activity of RA and in the development of ILD. Although the underlying mechanism remained not fully understood. Multiple studies have

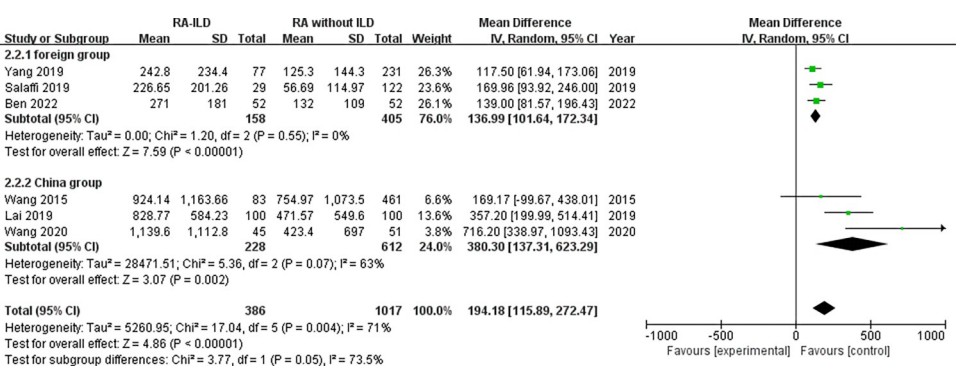

**Fig 14. Forest plot of subgroup analysis for the ACPA titer.**

demonstrated a high prevalence of both subclinical and clinical ILD throughout the RA disease course. Subclinical and clinical ILD occur frequently in preclinical, early, and established RA and may play a key role in RA-related autoantibody production and disease progression [32]. Multiple epidemiologic and genetic risk factors, as well as serum biomarkers, have been associated with RA-ILD. In this meta-analysis, the pooled result of 18 studies with dichotomous data showed that RF was significantly related to the risk of RA-ILD, heterogeneity across the studies was low. The combined effect of 5 studies with continuous data showed that the RF titer of RA-ILD group was significantly higher than that of RA without ILD group. Due to significantly heterogeneity for the analysis of RF titer (continuous data) and RA-ILD risk, a sensitivity analysis was performed to determine the reason for high heterogeneity. The pooled effect did not change significantly, and the risk estimates of RF titer and RA-ILD were stable. Sensitivity analysis revealed that "Salaffi 2019" primarily caused the heterogeneity. Meanwhile, the quantitative pooled results of ACPA and the qualitative pooled results of ACPA showed that there was significantly correlation between ACPA and RA-ILD risk. No significant heterogeneity was observed for dichotomous data of ACPA among the included 13 studies. Although the results of the pooled analysis showed that the high titers of ACPA were significantly related to the risk of RA-ILD, while considerable heterogeneity was detected. Sensitivity analysis revealed that the study by "Wang 2020" was likely to be the source of heterogeneity. When we omitted this study, the $I^2$ value dropped significantly from 71% to 52%. Furthermore, subgroup analysis was performed to determine the source of heterogeneity between the 6 studies of ACPA titer, subgroup analysis found no causes of heterogeneity. However, there was no heterogeneity was observed in the subgroup of foreign group (chi$^2$ = 1.20, $P$ = 0.55, $I^2$ = 0%; random effects model). Currently, the exact etiology of RA-ILD is not known. However, some factors and serum biomarkers have been reported to be involved in the pathogenesis of RA-ILD. It has been proposed that the lung is a production site for citrullinated proteins by peptidylarginine deiminases [33,34], some of these proteins may trigger immune responses in the lung and finally result in high titers of antibodies to citrullinated protein antigens. Recent studies found that citrullination is not only involved in the development of joint damage in RA but is also found in the bronchoalveolar lavage fluid of RA-associated interstitial pneumonia and idiopathic interstitial pneumonia subjects [35]. Based on these observations, two potential mechanisms that explain the pathogenesis of RA-ILD were proposed by Paulin et al [36]. First, citrullinated peptides of RA cause the activation and differentiation of lung fibroblasts into myofibroblasts to produce fibers. Second, aging alveolar epithelial cells promote the formation of fibers in susceptible patients. Environmental damage (such as smoking) leads to oxidative stress, which promotes fibroblast differentiation and proliferation. In addition, some studies demonstrated that lung abnormalities, increased protein citrullination, and ACPA enrichment in the lungs were present early after disease onset. This conjecture was strengthened by the finding that citrullinated proteins have been identified in lung tissue [35]. It is not clear whether these antibodies contribute to the initiation of pulmonary diseases in RA patients or whether they merely reflect ongoing pulmonary injury. Moreover, we identified that higher ESR and higher CRP were associated with RA-ILD. Heterogeneity between the seven studies of ESR was considerable, sensitivity analysis revealed that "Salaffi 2019" primarily caused the heterogeneity. When we omitted this study, the $I^2$ value dropped significantly from 52% to 27%. We also analyzed the association between positive ANA and RA-ILD risk, while our meta-analysis has shown no association between ANA positive status and risk of RA-ILD. Additionally, we assumed that publication bias would not be a big issue in our analysis. "Salaffi 2019" might introduce bias for the analysis of RF titer, when we omitted this study, the $P$-value of Egger's test increased significantly from 0.024 to 0.063. "Wang 2020" might introduce bias for the analysis of ACPA titer, when we omitted this study, the $P$-value of Egger's test increased

significantly from 0.044 to 0.204. Nevertheless, the *P*-value of Egger's test was 0.024 for the analysis of positive ACPA, indicating that significant publication bias was observed.

In this study, we conducted a meta-analysis to identify the factors associated with the development of RA-ILD. Our study has some limitations that should be considered. The data should be interpreted cautiously due to the limited number of high-quality studies and the limitations inherent in observational studies. First, the studies included were heterogeneous and had variable methodological quality. Moreover, most of the studies were retrospective in design, with small sample sizes and conducted in single medical institutions. Second, the number of patients enrolled, RA disease duration, age at onset of RA and population distribution varied across studies and significant heterogeneity of male, average age and age at onset of RA among studies was noted, which might be attributed to study design, population characteristics, regions and sample size. Although the total number of studies included was not small, more studies, especially prospective studies with large sample sizes, are still needed to overcome all of these limitations.

## Conclusions

In summary, our meta-analysis identified that male sex, older age, longer duration of RA, older age at onset of RA, smoking, positive RF, positive ACPA, elevated RF titer, elevated ACPA titer, higher ESR and higher CRP were factors associated with RA-ILD. Other factors contributing to heterogeneity may have been unidentified in our review.

## Supporting information

**S1 Table. Characteristics of the included studies.**
(DOC)

**S2 Table. Quality scores of the included studies.**
(DOC)

**S1 Appendix. PRISMA 2020 checklist.**
(PDF)

## Acknowledgments

We really appreciate the efforts of all the researchers whose articles were included in this study.

## Author Contributions

**Conceptualization:** Minjie Zhang.

**Data curation:** Minjie Zhang, Jianwei Yin, Xiaoyan Zhang.

**Formal analysis:** Minjie Zhang, Jianwei Yin, Xiaoyan Zhang.

**Funding acquisition:** Jianwei Yin.

**Methodology:** Minjie Zhang, Jianwei Yin, Xiaoyan Zhang.

**Software:** Minjie Zhang, Jianwei Yin.

**Supervision:** Jianwei Yin, Xiaoyan Zhang.

**Visualization:** Minjie Zhang.

**Writing – original draft:** Minjie Zhang, Jianwei Yin.

**Writing – review & editing:** Minjie Zhang, Jianwei Yin.

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
