## [Decision Letter · Decision Letter 0]

3 Apr 2023

PONE-D-23-05807Factors associated with interstitial lung disease in patients with rheumatoid arthritis: A systematic review and meta-analysisPLOS ONE

Dear Dr. Yin,

Thank you for submitting your manuscript to PLOS ONE. After careful consideration, we feel that it has merit but does not fully meet PLOS ONE’s publication criteria as it currently stands. Therefore, we invite you to submit a revised version of the manuscript that addresses the points raised during the review process.

We look forward to receiving your revised manuscript.

Kind regards,

Jan René Nkeck, M.D., M.Sc

Academic Editor

PLOS ONE

Journal Requirements:

https://link.springer.com/article/10.1007/s10067-021-05808-2?code=998b3470-2dc8-4ff0-98f5-cdc4b83846cd&error=cookies_not_supported

https://link.springer.com/article/10.1007/s10067-021-05808-2?code=998b3470-2dc8-4ff0-98f5-cdc4b83846cd&error=cookies_not_supported

In your revision ensure you cite all your sources (including your own works), and quote or rephrase any duplicated text outside the methods section. Further consideration is dependent on these concerns being addressed

3. We note that you have stated that you will provide repository information for your data at acceptance. Should your manuscript be accepted for publication, we will hold it until you provide the relevant accession numbers or DOIs necessary to access your data. If you wish to make changes to your Data Availability statement, please describe these changes in your cover letter and we will update your Data Availability statement to reflect the information you provide

Reviewers' comments:

Reviewer's Responses to Questions

**Comments to the Author**

1. Is the manuscript technically sound, and do the data support the conclusions?

Reviewer #1: Yes

Reviewer #2: Yes

Reviewer #3: Partly

2. Has the statistical analysis been performed appropriately and rigorously? 

Reviewer #1: Yes

Reviewer #2: Yes

Reviewer #3: Yes

3. Have the authors made all data underlying the findings in their manuscript fully available?

Reviewer #1: Yes

Reviewer #2: Yes

Reviewer #3: Yes

4. Is the manuscript presented in an intelligible fashion and written in standard English?

Reviewer #1: Yes

Reviewer #2: Yes

Reviewer #3: No

5. Review Comments to the Author

Reviewer #1: this is an interesting work that defines the RA population at risk of ILD and therefore requires special monitoring, however:-

- It is useful to define the risk factors for IPF,

-Analyze the population that presents the association ACPA and FR

-Some authors have found obstructive ventilatory deficit in ACPA patients (Zaccardelli, Ketfi) is what you have identified in your meta-analysis of these anomalies.

Reviewer #2: Minor revisions

Introduction :

• Lines 63- 64, “… and data extracted by two authors…” : maybe a missing word here, please check

• Line 65, “arthritis”,”rheumatoid” : maybe a missing “OR” /”AND” here, please check

Material and Methods

• Line 75, “Studies reporting….”: did the authors mean “reporting only….” since even studies reporting risk factors could also report a prevalence and even outcome

• Line 84, “apply to cohort and… “: did authors mean “applied”? Please check

Results

• Lines 114-115, “the identified….smoking”, this sentence seems to be incomplete, please reformulate

• Lines 116-117, “revealed that ……risk of RA-ILD” : redundant information, please reformulate

• Table 1 : first column, please insert ACPA before Foreign group and China group, since this division concerned only ACPA.

Discussion

• Lines 196-197 : “of…. ACPA titer, and cannot be explain” : not clear, use of present time at the end of sentence, while past tense is use at the beginning of the sentence. Please re formulate

Major revisions

Introduction

• Is RA-ILD or its worsening preventable? Authors should clearly state this, to support the assertion on lines 47-48 “with the goal of preventing irreversible damage”

Methodology

• Authors should state whether the evaluation of the quality was blind or open : did the authors assessing the quality have access to article data such as authors names, journal, etc?

• Authors should provide the repartition of studies by language (Chinese and English) in the S1 table

• Figure 1 : on the first two steps of the flow chart, reasons for withdrawal are missing

Discussion

• An important question remains unresolved : are the factors you found predictors/risk factors of ILD only, or deal with RA severity. The clinical severity of RA (in terms of number and intensity of involved joints, as well as other extra articular involvement) has not been mentioned. Is it known as a factor associated with RA-ILD, if yes, this would be a major confounder. In fine, Is ILD just a marker of RA severity? Authors should provide data on severity, or at least (if data not available) discuss this issue.

Conclusion

• Authors should replace “risk factors of” by “factors associated with” as said in the title, since incident ILD have not been studied and the causality cannot be demonstrated from the data used

Reviewer #3: 1. The title and objective of the study do not tie as factors associated with a disease do not always imply they are risk factors.

2. In the methodology section, we have difficulty understanding certain points as they appear unclear us. These include

- The search strategy focused on studies published in English and Chinese, thus rendering studies carried out in other languages e.g. French to not be included. This could lead to loss of information as not all properly conducted studies are included.

- The exclusion criteria "data could not be extracted". Was it because of the lack of a full text article or language barrier or other factor that made the data not to be able to be extracted? We believe this point should be made clearer.

- The quality assessment of the studies is incomplete to us as we see scores of 5,6, 8 etc. Nothing is mentioned after this score to permit us know if the quality of the included studies were good enough to ensure we can trust the results.

3. In the results section:

- The figures presented are not on their own able to inform a review on what is presented because they have no titles and no clear legends

- The authors state presence of significant heterogeneity in the studies on several aspects yet still go on to have pooled estimates on these and even reach conclusions with some of these variables. This to us is of some concern.

- There also is risk of publication bias for several factors which later are considered as "risk factors" for RA-ILD. This to us is of some concern.

In general, there are many grammatical and typographical errors which make it difficult to read and understand the manuscript.

6. PLOS authors have the option to publish the peer review history of their article (what does this mean?). If published, this will include your full peer review and any attached files.

Reviewer #1: No

Reviewer #2: No

Reviewer #3: No

---

## [Author Response · Author response to Decision Letter 0]

11 Apr 2023

Dear Editors and Reviewers:

Thank you for your letter and for the reviewers’comments concerning our manuscript entitled “Factors associated with interstitial lung disease in patients with rheumatoid arthritis: A systematic review and meta-analysis” (ID: PONE-D-23-05807).Those comments are all valuable and very helpful for revising and improving our paper, as well as the important guiding significance to our researches. We have studied comments carefully and have made correction which we hope meet with approval. The main corrections in the paper and the responds to the reviewer’s comments are as flowing:

Responds to the reviewer’s comments:

Reviewer #1: this is an interesting work that defines the RA population at risk of ILD and therefore requires special monitoring, however:-

- It is useful to define the risk factors for IPF

1. IPF (Idiopathic Pulmonary Fibrosis) is defined as a specific form of chronic, progressive fibrosing interstitial pneumonia of unknown cause, occurring primarily in older adults. The definition of IPF requires the exclusion of other forms of interstitial pneumonia including other idiopathic interstitial pneumonias and ILD associated with environmental exposure, medication, or systemic disease. There are a number of clinical and mechanistic parallels between IPF and other fibrosing ILDs that may present a progressive phenotype. 

2. The diagnosis of IPF requires: Exclusion of other known causes of interstitial lung disease (ILD) (e.g., domestic and occupational environmental exposures, connective tissue disease, and drug toxicity). Although IPF is, by definition, a disease of unknown etiology, a number of potential risk factors have been described such as: cigarette smoking, environmental exposures, microbial agents, gastroesophageal reflux and genetic factors. There are no reliable data on the role of screening serologies in patients with suspected IPF [1]. Park et al [2] conducted a systematic review and meta-analysis to evaluate the risk factor of IPF. They found that metal dust, wood dust, pesticide, occupational history of farming or agriculture and ever smoking increased the risk of IPF. It is really true as Reviewer mentioned that cigarette smoking, older age and environmental exposures were risk factors for IPF. The diagnosis of IPF requires exclusion of other known causes of ILD, such as connective tissue diseases (CTDs), encompass a spectrum of systemic immune diseases in which self-reactive T- and B-cells produce circulating autoantibodies leading to inflammation and organ damage. CTDs associated with ILD include RA, SSc, idiopathic inflammatory myopathy (polymyositis and dermatomyositis), Sjögren's syndrome, systemic lupus erythematosus and mixed CTDs. Of these conditions, SSc and RA are most commonly associated with ILD [3]. There are strong associations between RA-ILD and older age, male sex, cigarette smoking, high rheumatoid factor titres and increased anti-citrullinated protein antibody levels. However, there are a few patients who were defined as IPF can detect high titer of rheumatoid factor, anti-cyclic citrullinated peptide. Patients with IPF may have a mildly positive antinuclear antibody titer and/or rheumatoid factor level. 

[1] Raghu G, Collard HR, Egan JJ, Martinez FJ, Behr J, Brown KK, et al. An official ATS/ERS/JRS/ALAT statement: idiopathic pulmonary fibrosis: evidence-based guidelines for diagnosis and management. American journal of respiratory and critical care medicine. 2011;183(6):788-824. https://doi.org/10.1164/rccm.2009-040GL PMID: 21471066

[2] Park Y, Ahn C, Kim TH. Occupational and environmental risk factors of idiopathic pulmonary fibrosis: a systematic review and meta-analyses. Scientific reports. 2021;11(1):4318. https://doi.org/10.1038/s41598-021-81591-z PMID: 33654111

[3] Cottin V, Hirani NA, Hotchkin DL, Nambiar AM, Ogura T, Otaola M, et al. Presentation, diagnosis and clinical course of the spectrum of progressive-fibrosing interstitial lung diseases. European respiratory review : an official journal of the European Respiratory Society. 2018;27(150). https://doi.org/10.1183/16000617.0076-2018 PMID: 30578335

-Analyze the population that presents the association ACPA and FR

All the data in this systematic review and meta-analysis were derived from included 22 articles. We did not recruit patients. The detailed information of population is not available to us. We are very sorry for that.

-Some authors have found obstructive ventilatory deficit in ACPA patients (Zaccardelli, Ketfi) is what you have identified in your meta-analysis of these anomalies.

We found these studies as follows:

[4] Zaccardelli A, Liu X, Ford JA, Cui J, Lu B, Chu SH, et al. Elevated Anti-Citrullinated Protein Antibodies Prior to Rheumatoid Arthritis Diagnosis and Risks for Chronic Obstructive Pulmonary Disease or Asthma. Arthritis care & research. 2021;73(4):498-509. https://doi.org/10.1002/acr.24140 PMID: 31961487 

As we can see from the Zaccardelli statements: ACPA positivity prior to RA onset was significantly associated with increased COPD risk, particularly in the pre-RA period. This may be due to loss of immune tolerance in the lungs from increased citrullination and ACPA production occurring before RA onset that pre-disposes to chronic airway disease and not explained by smoking. Women who went on to develop RA were also more likely to develop asthma compared to matched controls, however this was risk was not related to pre-RA ACPA status. 

[5] Ketfi A, Tahiat A, Djouadi C, Djenouhat K, Ben Saad H. Lung function data of North-African patients with rheumatoid arthritis: a comparative study between anti-citrullinated peptides antibodies positive and negative patients. La Tunisie medicale. 2022;100(8-9):626-41. PMID: 36571731

Methods: This comparative pilot study was performed over a two-year period (2018-2019) in Algiers (Algeria). The study included two groups of RA non-smoker patients: 26 ACPA+ and 33 ACPA-. RA was diagnosed according to the ACR/EULAR 2010 RA classification criteria. Spirometry and plethysmography were performed. The following definitions were applied: Obstructive Ventilatory Impairment (OVI): FEV1/FVC z-score < -1.645; Restrictive Ventilatory Impairment (RVI): Total Lung Capacity (TLC) z-score < -1.645; Mixed Ventilatory Impairment (MVI): FEV1/FVC z-score < -1.645 and TLC z-score < -1.645; lung- hyperinflation: residual volume z-score > +1.645; Nonspecific Ventilatory Impairment (NSVI): FEV1 z-score < -1.645, FVC z-score < -1.645, FEV1/FVC z-score ≥ -1.645, and TLC z-score ≥ -1.645.

1. As we can see from the Ketfi statements, the study included two groups of RA patients: 26 ACPA+ and 33 ACPA-.RA is a systemic inflammatory disorder, with the most common extra-articular manifestation of RA being lung involvement. While essentially any of the lung compartments can be affected and manifest as interstitial lung disease (ILD), pleural effusion, cricoarytenoiditis, constrictive or follicular bronchiolitis, bronchiectasis, pulmonary vasculitis, and pulmonary hypertension, RA-ILD is a leading cause of death in patients with RA and is associated with significant morbidity and mortality. Though lung involvement in RA typically occurs following articular manifestations, pulmonary manifestations may occasionally precede joint symptoms [6]. Several studies have shown that pulmonary function testing (PFT) is helpful for the early detection of RA-ILD. PFT in RA-ILD patients often reveals restrictive ventilatory defects with decreases in gas exchange, and such abnormalities may be detected even in the absence of any clinical symptoms [7].

2. Prevalence of airway disease in RA is high and can occur in 39–60% of patients. Both the large (upper and lower) and distal small airways can be involved. The most common manifestations are bronchiectasis, bronchiolitis, airway hyperreactivity and cricoarytenoid arthritis[6]. As we can see from the Zaccardelli statements: ACPA positivity prior to RA onset was significantly associated with increased COPD risk, particularly in the pre-RA period. It has been found that patients with RA typically have circulating antibodies, most notably RF and ACPAs. These autoantibodies are present in an estimated 50–80% of RA patients and are often (though not always) present in the serum for several years prior to clinical disease onset. In individuals who are genetically predisposed, citrullination, a post-translational modification marked by the conversion of arginine to citrulline, can trigger an immune response leading to the production of ACPAs, which are strongly associated with development of RA.

3. Overall, Zaccardelli and Ketfi emphasize that ACPA positivity was significantly associated with increased obstructive ventilatory deficit risk, these results emphasize that clinicians should closely monitor seropositive RA patients for airway abnormalities, even prior to RA onset. Future work is needed to elucidate the bi-directional link between RA and chronic airway diseases.

[6] Kadura S, Raghu G. Rheumatoid arthritis-interstitial lung disease: manifestations and current concepts in pathogenesis and management. European respiratory review : an official journal of the European Respiratory Society. 2021;30(160). https://doi.org/10.1183/16000617.0011-2021 PMID: 34168062

[7] Solomon JJ, Chung JH, Cosgrove GP, Demoruelle MK, Fernandez-Perez ER, Fischer A, et al. Predictors of mortality in rheumatoid arthritis-associated interstitial lung disease. The European respiratory journal. 2016;47(2):588-96. https://doi.org/10.1183/13993003.00357-2015 PMID: 26585429

Reviewer #2: Minor revisions

Introduction :

• Lines 63- 64, “… and data extracted by two authors…” : maybe a missing word here, please check

Searches were restricted to articles written in English and Chinese. We searched all relevant studies on RA-ILD published before December 30, 2022 and two authors extracted data from articles independently.

• Line 65, “arthritis”,”rheumatoid” : maybe a missing “OR” /”AND” here, please check

“rheumatoid arthritis” OR “arthritis” OR “rheumatoid” 

Material and Methods

• Line 75, “Studies reporting….”: did the authors mean “reporting only….” since even studies reporting risk factors could also report a prevalence and even outcome

studies reporting only

• Line 84, “apply to cohort and… “: did authors mean “applied”? Please check

The methodological quality of cohort and case-control studies were assessed using the Newcastle-Ottawa scale (NOS) with a total score of 9 points.

Results

• Lines 114-115, “the identified….smoking”, this sentence seems to be incomplete, please reformulate

The identified factors associated with RA-ILD include average age, duration of RA, age at onset of RA and smoking.

• Lines 116-117, “revealed that ……risk of RA-ILD” : redundant information, please reformulate

revealed that older age was associated with RA-ILD.

• Table 1 : first column, please insert ACPA before Foreign group and China group, since this division concerned only ACPA.

Discussion

• Lines 196-197 : “of…. ACPA titer, and cannot be explain” : not clear, use of present time at the end of sentence, while past tense is use at the beginning of the sentence. Please re formulate

Furthermore, subgroup analysis was performed to determine the source of heterogeneity between the 6 studies of ACPA titer, subgroup analysis found no causes of heterogeneity.

Major revisions

Introduction

• Is RA-ILD or its worsening preventable? Authors should clearly state this, to support the assertion on lines 47-48 “with the goal of preventing irreversible damage”

To date, there have not been any guidelines or RCTs for the treatment of RA-ILD. From recent data, conventional disease-modifying anti-rheumatic drugs (cDMARDS), including Methotrexate (MTX) and Leflunomide (LEF), play beneficial roles in the prevention and treatment of RA-ILD. Biologic disease-modifying anti-rheumatic drugs (bDMARDs) such as abatacept and rituximab are effective in stabilizing ILD involvement in RA patients [8]. Although a great deal of research has been conducted and breakthroughs have been achieved in the field of RA, there are still many questions about RA-ILD. How do clinicians help RA patients prevent the onset of ILD? When should clinicians initiate the treatment and choose the right option for different ILD patterns? However, most approaches are based on experts, without evidence from RCTs. To effectively care for these patients, further research is needed to define the pathogenesis and treatment of RA-ILD.

[8] Dai Y, Wang W, Yu Y, Hu S. Rheumatoid arthritis-associated interstitial lung disease: an overview of epidemiology, pathogenesis and management. Clinical rheumatology. 2021;40(4):1211-20. https://doi.org/10.1007/s10067-020-05320-z PMID: 32794076

Methodology

• Authors should state whether the evaluation of the quality was blind or open: did the authors assessing the quality have access to article data such as authors names, journal, etc?

the methodological quality of cross-sectional studies was evaluated using the assessment involving 11 items recommended by the Agency for Healthcare Research and Quality (AHRQ). The total score ranged from 0 to 11, with a score of 8 or higher considered high quality. The methodological quality of cohort and case-control studies were assessed using the Newcastle-Ottawa scale (NOS) with a total score of 9 points. A total score of 5 or less was considered low, 6 or 7was considered moderate, and 8 or 9 was deemed high quality. four of the studies were considered low quality, sixteen were considered moderate quality, and two were determined to be high quality based on their total scores. Evaluation of the quality of included studies was performed by 2 authors independently based on the study design. Two independent authors were blinded to the authors names, titles and years of publication of the included studies. Discrepancies in scores were resolved by consensus with a third author. The methodological quality assessment results were shown in S2 Table.

• Authors should provide the repartition of studies by language (Chinese and English) in the S1 table

We have made correction according to the Reviewer’s comments. 

• Figure 1 : on the first two steps of the flow chart, reasons for withdrawal are missing

We have made correction according to the Reviewer’s comments.

Discussion

• An important question remains unresolved : are the factors you found predictors/risk factors of ILD only, or deal with RA severity. The clinical severity of RA (in terms of number and intensity of involved joints, as well as other extra articular involvement) has not been mentioned. Is it known as a factor associated with RA-ILD, if yes, this would be a major confounder. In fine, Is ILD just a marker of RA severity? Authors should provide data on severity, or at least (if data not available) discuss this issue.

We have re-written this part according to the Reviewer’s suggestion

1. This systematic review and meta-analysis were performed to identify risk factors for RA-ILD，

2. Whether the clinical severity of RA (in terms of number and intensity of involved joints, as well as other extra articular involvement) increases the risk of RA-ILD is disputed. We aimed to evaluate the association of demographic characteristics and laboratory items with development of RA-ILD. 

3. The latest research shows that RA-ILD is a serious extra-articular complication of RA that involves several radiologic and pathologic subtypes. Previously considered a consequence of prolonged disease severity in longstanding RA, subclinical and clinical ILD are increasingly recognized throughout the entire RA disease course. Multiple studies have demonstrated a high prevalence of both subclinical and clinical ILD throughout the RA disease course. A significant proportion of RA-ILD patients develop ILD prior to articular manifestations, suggesting that the lung plays a central role RA development, perhaps through ACPA production. RA-ILD also occurs in early RA, when exuberant autoantibody production and systemic inflammation may propagate disease activity [9]. 

[9] McDermott GC, Doyle TJ, Sparks JA. Interstitial lung disease throughout the rheumatoid arthritis disease course. Current opinion in rheumatology. 2021;33(3):284-91. https://doi.org/10.1097/bor.0000000000000787 PMID: 33625044

Conclusion

• Authors should replace “risk factors of” by “factors associated with” as said in the title, since incident ILD have not been studied and the causality cannot be demonstrated from the data used

We have made correction according to the Reviewer’s comments.

Reviewer #3: 1. The title and objective of the study do not tie as factors associated with a disease do not always imply they are risk factors.

We have made correction according to the Reviewer’s comments.

2. In the methodology section, we have difficulty understanding certain points as they appear unclear us. These include

- The search strategy focused on studies published in English and Chinese, thus rendering studies carried out in other languages e.g. French to not be included. This could lead to loss of information as not all properly conducted studies are included.

We are very sorry for our negligence of studies carried out in other languages. Due to the limitation of language, we failed to include more research. We look forward to more studies carry out in English.

- The exclusion criteria "data could not be extracted". Was it because of the lack of a full text article or language barrier or other factor that made the data not to be able to be extracted? We believe this point should be made clearer.

The studies which data could not be extracted as follows:

[10] Natalini JG, Baker JF, Singh N, Mahajan TD, Roul P, Thiele GM, et al. Autoantibody Seropositivity and Risk for Interstitial Lung Disease in a Prospective Male-Predominant Rheumatoid Arthritis Cohort of U.S. Veterans. Annals of the American Thoracic Society. 2021;18(4):598-605. https://doi.org/10.1513/AnnalsATS.202006-590OC PMID: 33026891. In this study, the patients with combined RF/ACPA seropositivity had a higher probability of prevalent ILD compared with seronegative subjects. RF titers demonstrated a monotonic association with prevalent ILD (OR, 2.69; 95% CI, 1.11–6.51 for low-positive [15–45 IU/ml] titers; OR, 3.40; 95% CI, 1.61–7.18 for high-positive [>45 IU/ml] titers; P for trend 0.01). Patients with high-positive (>15 U/ml) ACPA titers were also at higher risk for prevalent ILD (OR, 1.91; 95% CI, 1.04–3.49) compared with ACPA-negative subjects. Combined RF/ACPA seropositivity was not associated with increased risk for incident ILD, nor were high- or low-positive RF or ACPA titers. It is impossible for us to extract data.

[11] Inui N, Enomoto N, Suda T, Kageyama Y, Watanabe H, Chida K. Anti-cyclic citrullinated peptide antibodies in lung diseases associated with rheumatoid arthritis. Clinical biochemistry. 2008;41(13):1074-7. https://doi.org/10.1016/j.clinbiochem.2008.06.014 PMID: 18638466. In this study, values are expressed as the number or median (interquartile range)

[12] Huang S, Doyle TJ, Hammer MM, Byrne SC, Huang W, Marshall AA, et al. Rheumatoid arthritis-related lung disease detected on clinical chest computed tomography imaging: Prevalence, risk factors, and impact on mortality. Seminars in arthritis and rheumatism. 2020;50(6):1216-25. https://doi.org/10.1016/j.semarthrit.2020.08.015 PMID: 33059295. In this study, RA-related lung disease was defined as ILD, bronchiectasis, or pleural disease on the impression of the radiologic report. It is difficult to make a distinction of data that is related to ILD among the three types of lung disease.

- The quality assessment of the studies is incomplete to us as we see scores of 5,6, 8 etc. Nothing is mentioned after this score to permit us know if the quality of the included studies were good enough to ensure we can trust the results.

We have made correction according to the Reviewer’s comments. 

the methodological quality of cross-sectional studies was evaluated using the assessment involving 11 items recommended by the Agency for Healthcare Research and Quality (AHRQ). The total score ranged from 0 to 11, with a score of 8 or higher considered high quality. The methodological quality of cohort and case-control studies were assessed using the Newcastle-Ottawa scale (NOS) with a total score of 9 points. A total score of 5 or less was considered low, 6 or 7was considered moderate, and 8 or 9 was deemed high quality. four of the studies were considered low quality, sixteen were considered moderate quality, and two were determined to be high quality based on their total scores. Evaluation of the quality of included studies was performed by 2 authors independently based on the study design. Two independent authors were blinded to the authors names, titles and years of publication of the included studies. Discrepancies in scores were resolved by consensus with a third author. The methodological quality assessment results were shown in S2 Table.

3. In the results section:

- The figures presented are not on their own able to inform a review on what is presented because they have no titles and no clear legends

We are very sorry for this comment. We prepared the manuscript according to the MANUSCRIPT BODY FORMATTING GUIDELINES of PLOS.

- The authors state presence of significant heterogeneity in the studies on several aspects yet still go on to have pooled estimates on these and even reach conclusions with some of these variables. This to us is of some concern.

1. It is really true as reviewer suggested that significant heterogeneity exist in the studies. In order to identify possible sources of heterogeneity, we did an analysis to check the effect size and heterogeneity with each study removed one at a time. Sensitivity analysis was conducted to explore for potential outliers statistically and subgroup analysis was conducted to explain some of the potential sources of heterogeneity. Our study has some limitations that should be considered. It is possible that limitations exist in the searching and screening approach used, leading to eligible studies being inadvertently left out of the review. As the reviewer mentioned that the search strategy focused on studies published in English and Chinese, thus rendering studies carried out in other languages e.g. French to not be included.

2. The number of patients enrolled, RA disease duration, age at onset of RA and population distribution varied across studies and significant heterogeneity of male, average age and age at onset of RA among studies was noted, which might be attributed to study design, population characteristics, regions and sample size. The specific reasons for the heterogeneity cannot be explained in our systematic review and meta-analysis.

- There also is risk of publication bias for several factors which later are considered as "risk factors" for RA-ILD. This to us is of some concern.

Publication bias is the tendency of studies to be more likely to be published when they report positive and statistically significant results, or have larger effect sizes. 22 articles were included in our systematic review and meta-analysis, but not every study contains all the factors. The publication bias exists only partially.

 In general, there are many grammatical and typographical errors which make it difficult to read and understand the manuscript.

We have made correction according to the Reviewer’s comments.

---

## [Decision Letter · Decision Letter 1]

11 May 2023

Factors associated with interstitial lung disease in patients with rheumatoid arthritis: A systematic review and meta-analysis

PONE-D-23-05807R1

Dear Dr. Yin,

We’re pleased to inform you that your manuscript has been judged scientifically suitable for publication and will be formally accepted for publication once it meets all outstanding technical requirements.

Kind regards,

Jan René Nkeck, M.D., M.Sc

Academic Editor

PLOS ONE

Reviewers' comments:

Reviewer's Responses to Questions

**Comments to the Author**

1. If the authors have adequately addressed your comments raised in a previous round of review and you feel that this manuscript is now acceptable for publication, you may indicate that here to bypass the “Comments to the Author” section, enter your conflict of interest statement in the “Confidential to Editor” section, and submit your "Accept" recommendation.

Reviewer #1: All comments have been addressed

Reviewer #2: All comments have been addressed

2. Is the manuscript technically sound, and do the data support the conclusions?

Reviewer #1: Yes

Reviewer #2: (No Response)

3. Has the statistical analysis been performed appropriately and rigorously? 

Reviewer #1: Yes

Reviewer #2: (No Response)

4. Have the authors made all data underlying the findings in their manuscript fully available?

Reviewer #1: Yes

Reviewer #2: (No Response)

5. Is the manuscript presented in an intelligible fashion and written in standard English?

Reviewer #1: Yes

Reviewer #2: (No Response)

6. Review Comments to the Author

Reviewer #1: (No Response)

Reviewer #2: (No Response)

7. PLOS authors have the option to publish the peer review history of their article (what does this mean?). If published, this will include your full peer review and any attached files.

Reviewer #1: No

Reviewer #2: No

---

## [Editor Report · Acceptance letter]

14 Jun 2023

PONE-D-23-05807R1 

Factors associated with interstitial lung disease in patients with rheumatoid arthritis: A systematic review and meta-analysis 

Dear Dr. Yin:

I'm pleased to inform you that your manuscript has been deemed suitable for publication in PLOS ONE. Congratulations! Your manuscript is now with our production department. 

Kind regards, 

on behalf of

Dr. Jan René Nkeck 

Academic Editor

PLOS ONE